# Sensitivity and specificity of Dried Blood Spot and Plasma Separation Card samples for Hepatitis C Virus RNA Testing

Agnes Malobela[1]*, Marie Amougou-Atsama[2], Panagiotis Iliopoulos[3],
Jean-Claude Mugisha[4], Nino Berishvili[5], Manana Sologashvili[6], Emmanuel Fajardo[1],
Francois Lamoury[1], Aurélien Macé[1], Maxwell Chirehwa[7], Richard Njouom[2],
Angelos Hatzakis[3], Jules Kabahizi[4], Claude Mambo Muvunyi[8], Penny Buxton[9],
Sadaf Mohiuddin[9], Maia Alkhazashvili[5], Elena Ivanova Reipold[1]*

**1** FIND, the global alliance for diagnostics, Geneva, Switzerland, **2** Centre Pasteur du Cameroun,
Yaoundé, Cameroon, **3** Hellenic Scientific Society for the Study of AIDS, Sexually Transmitted and
Emerging Diseases, Athens, Greece, **4** Rwanda Military Hospital, Kigali, Rwanda, **5** National Center for
Disease Control and Public Health, Tbilisi, Georgia, **6** Hepa Plus, Tbilisi, Georgia, **7** University of Cape
Town, Cape Town, South Africa, **8** Rwanda Biomedical Center, Kigali, Rwanda, **9** National Reference
Laboratory, Victoria, Australia

\* agnes.malobela@finddx.org (AM), elena.reipold@protonmail.com (EIR)

pgph.0006082

CANADA

**Peer Review History:** PLOS recognizes the
benefits of transparency in the peer review
process; therefore, we enable the publication
of all of the content of peer review and
author responses alongside final, published
articles. The editorial history of this article is
available here: https://doi.org/10.1371/journal.
pgph.0006082

## Abstract

Dried blood spots (DBS) and plasma separation cards (PSC) have the potential to
improve access to hepatitis C virus (HCV) testing. This multicenter study evaluated
the performance of two HCV RNA assay platforms using capillary or venous DBS and
PSC.

Participants were enrolled in Cameroon, Rwanda, Georgia, and Greece. DBS and
PSC prepared from capillary and venous blood samples were collected from three
target populations; individuals at risk of HCV infection, persons living with HCV, and
individuals previously treated for HCV. The diagnostic accuracy of DBS and PSC for
detecting HCV RNA was assessed using the cobas HCV nucleic assay on the cobas
4800 and cobas 6800 platforms (Roche), with plasma samples tested using these
platforms serving as the reference standard.

A total of 936 participants were enrolled. Capillary, venous DBS and venous PSC
demonstrated high diagnostic accuracy. Sensitivity and specificity were 97.2% (95%
CI: 95.2–98.3) and 88.6% (95%CI: 85.4 –91.2) for capillary DBS, 96.1% (95%CI:
93.9 –97.5) and 87.8% (95%CI: 85.4 –90.4) for venous DBS and 95.2% (95%CI:
92.8 –96.8) and 99.6% (95%CI: 98.5 –99.9) for venous PSC, on the cobas 4800. The
cobas 6800 platform had a sensitivity and specificity of 97.3% (95%CI: 95.4 –98.5)
and 95.9% (95%CI: 93.7 –97.3) on venous DBS, 96.7% (95%CI: 94.6 –98.0) and
99.8% (95%CI: 98.8 – 100) on venous PSC, and 96.9% (95%CI: 94.8 –98.1) and

which permits unrestricted use, distribution, and reproduction in any medium, provided the original author and source are credited.

**Data availability statement:** The datasets analysed during the current study are available in the Figshare repository. Where applicable, data-sharing agreements with participating countries were established. https://figshare.com/articles/dataset/Roche_DBS_and_Plasma_Separation_Data_Set_-_HCV_RNA_Detection/29543879.

**Funding:** This research was funded by UNITAID, grant number UA-HCV01. The funder had no role in study design, data collection and analysis, decision to publish, or preparation of the manuscript. Roche Diagnostics provided the reagents for testing at the Australian site - National Reference Laboratory, Victoria, Australia. FIND staff; EIR, FMJL, AM (Agnes Malobela), AM (Aurélien Mace), EF were funded through UNITAID, grant number UA-HCV01.

**Competing interests:** To the best of our search in publicly available literature and patent records, Roche currently markets HCV RNA detection assays for the cobas® 4800, 5800, 6800, and 8800 platforms. The cobas® Plasma Separation Card is a commercially available sample collection device that is associated with patents covering its design and use for plasma separation. The regulatory approval status of the Plasma Separation Card specifically for HCV RNA testing varies by region. We are not aware of any additional patented products, in development diagnostics, or separate marketed products beyond these associated Roche assays and the plasma separation card in connection with HCV RNA detection. We confirm that the funding and the above competing interests do not alter our adherence to PLOS Global Public Health policies on the sharing of data and materials.

99.8% (95%CI: 98.8 – 100) on capillary PSC. The diagnostic accuracy of capillary and venous DBS and PSC for detecting HCV RNA was high on the two platforms evaluated.

This study demonstrates that DBS and PSC sample types can be an alternative to plasma to screen for HCV infection, thus facilitating access to testing.

## Importance

Although the use of point-of-care rapid antibody tests for hepatitis C virus (HCV) is increasing, people who test positive must still undergo a confirmatory test before receiving treatment. Confirmatory testing for HCV RNA can be performed using plasma, serum, or capillary whole blood, depending on the available diagnostic platforms and testing algorithms. Plasma samples are more commonly used; however, their collection, storage, transport, and handling can pose significant challenges in low-resource settings. To improve access to HCV testing, there is a need for alternative sample types. Blood samples dried on filter paper cards (dried blood spots, or DBS) can be collected at the community level, have low biohazard risk, can withstand exposure to ambient temperatures, and are easily stored and shipped. Plasma Separation Cards (PSC), which collect plasma rather than whole blood, have similar properties. This study demonstrated that the diagnostic accuracy of DBS and PSC for detecting HCV RNA using the Roche cobas 4800 and 6800 platforms was high, confirming that they could be viable alternatives to plasma sampling for HCV testing.

## Introduction

An estimated 58 million people had chronic hepatitis C virus (HCV) infection worldwide in 2019 [1]. Due to limited access to HCV screening and diagnosis services, many persons living with HCV living in low- and middle-income countries (LMICs) are unaware of their status [2,3]. As part of efforts to tackle access to HCV testing barriers, the decentralized use of rapid HCV antibody (Ab) testing has increased in recent years [4]. Nevertheless, any person receiving an HCV Ab positive (Ab+) test result must receive an HCV RNA confirmation test. There are existing confirmatory tests that may use capillary blood for HCV confirmation testing such as the GeneXpert Cepheid assay [5–7]. This requires infrastructure to collect, store, transport, and handle plasma samples, as well as laboratories manned by highly-qualified technicians and equipped with technology to run nucleic acid amplification tests (NAAT) [1]. Hence, in some LMICs, HCV RNA testing might be available only in major cities and centralized laboratories.

Production of scientific evidence to simplify HCV diagnostic algorithms is paramount for further decentralization of HCV testing and integration with other routine care provision, with the aim of reaching global targets for HCV elimination by 2030

[4,8]. To simplify testing, alternatives to plasma samples are required. Whole blood collected and dried in a filter paper card (i.e., Dried Blood Spots or DBS) is a viable sampling alternative, as DBS can be collected at community level, have low biohazard risk, can withstand prolonged exposure to ambient temperatures, and can be easily stored and shipped to NAAT-equipped testing sites [9]. The use of cobas Plasma Separation Cards (PSC) (Roche, Switzerland) is another alternative. PSC share many features with DBS, but they collect plasma rather than whole blood. PSC may help overcome certain limitations of DBS such as interference from pro-viral DNA and intracellular RNA in whole blood, and inadequate lower limits of detection due to suboptimal volumes of viral RNA [10].

The diagnostic accuracy of various HCV RNA tests performed on DBS has been evaluated [11–13]. A systematic review of HCV RNA test performance from DBS samples demonstrated a pooled sensitivity of 98% and specificity of 98% [11]. Catlett et al (2022) also reported higher sensitivity and specificity for HCV RNA detection using DBS, particularly when analyses were restricted to samples above the limit of quantification [14].

The performance of PSC for HCV RNA and serological markers detection was demonstrated as comparable to that of conventional plasma and serum samples [12,15]. The utility of PSC for the determination of serological markers for HCV and for hepatitis B virus (HBV) among rural and economically-disadvantaged populations has been reported [13,16]. However, many of these diagnostic accuracy studies have been conducted at a single center and have not considered the impact of handling DBS or PSC at ambient temperatures.

To assess how viable DBS and PSC are as alternative sampling methods to increase HCV testing uptake by populations in LMICs, there is a need to consider the effect of the environment from the point of sample collection to the point of testing. To the best of our knowledge no studies have evaluated both DBS and PSC concurrently, however both DBS and PSC are considered in the literature as plausible alternatives for HCV RNA detection [12,17,18]. We conducted a multi-center study in four LMICs with differing HCV epidemiological profiles with the aim of evaluating the diagnostic accuracy of the cobas 4800 and cobas 6800 HCV RNA testing assays (Roche) performed on DBS and PSC samples transported at ambient temperatures, when compared with the standard HCV RNA method using plasma samples. This study was designed to use manufacturer-validated protocols for collection, handing and use of DBS and PSC samples and generated data contributed to the World Health Organization (WHO) updated recommendations on HCV simplified delivery and diagnostics [1].

## Materials and methods

### Study design

This diagnostic accuracy study was designed, conducted and reported in accordance with the Standards for Reporting of Diagnostic Accuracy Studies (STARD) guidelines [19]. This study was registered on clinicaltrials.gov (NCT03896087), and it was part of a large-scale evaluation of the accuracy of different HCV RNA assays and platforms using DBS. The clinical study was initially registered on 14 March 2019 on ClinicalTrials.gov with Registration number NCT03896087 available here: Study Details | Evaluation of Dried Blood Spot for HCV RNA Testing | ClinicalTrials.gov.

### Inclusivity in global research

Additional information regarding the ethical, cultural, and scientific considerations specific to inclusivity in global research is included in the Supporting Information (**S1 Checklist).**

### Ethics approval and consent to participate

The study was conducted in accordance with the Declaration of Helsinki, and approved by institutional review boards in Cameroon, Comite National d'ethique de la recherche pour la sante Humaine (CNERSH) - (N°2019/04/1157/CE/ CNERSH/SP), Georgia, National Centre for Disease Control and Public Health Institutional Review Board - (IRB#

2018–047 and IRB# 2019–019 for Protocol N°8160–2/1), Greece, The Institutional Review Board of the Hellenic Scientific Society for the Study of AIDS and Sexually Transmitted Diseases (EEEEAIDS) - (Protocol N°121) and Rwanda by the Rwanda National Ethics Committee (RNEC) - (N°121/RNEC/2020 and N°328/RNEC/2019). Written informed consent was obtained from all participants involved in the study.

### Study period and sites

The study was conducted between May 2019 and October 2020 in four sites located in four countries with differing HCV prevalence. These were: the Centre Pasteur du Cameroun (CPC) in Yaoundé (Cameroon) with recruitment beginning on 17 June 2019 and ending on 18 Oct 2019; HEPA Plus, a harm reduction center in Tbilisi (Georgia) with recruitment beginning on 07 May 2019 and ending on 25 Sep 2019; the Hellenic Scientific Society for the Study of AIDS and Sexually Transmitted Diseases (EEEEAIDS) in Athens (Greece) with recruitment beginning on 05 Sep 2019 and ending on 17 Dec 2019; the Rwanda Military Hospital (RMH) in Kigali (Rwanda) with recruitment beginning on 05 April 2020 and ending on 24 Sep 2020. The countries were selected according to their potential capacity for scaling up DBS protocols. Site selection considered (i) the use of high-throughput molecular platforms already in use for HIV viral load testing, along with the overall capacity of the system to absorb additional volumes and (ii) governmental readiness to adopt DBS protocols off-label before regulatory approval.

CPC and RMH primarily serve the general population, whereas Hepa Plus and EEEEAIDS provide services to people who inject drugs (PWID). Sample collection was performed by trained laboratory technicians (Cameroon and Rwanda) or by trained nurses and doctors in community-based harm reduction centres (Georgia, Greece).

### Study population and sample size

Three groups were included:

- '**HCV at-risk**': Adults with past and/or current exposure to risk factors, as per WHO [20] and Centres for Disease Control and Prevention (CDC) [21] guidelines.

- '**Anti-HCV Ab+**': Adults with a documented positive anti-HCV antibody serology result who had not initiated HCV treatment.

- '**HCV treated**': Adults previously treated with direct-acting antiviral (DAA) therapy and attending follow-up or test of cure visits.

A total sample of >815 participants was calculated to achieve 85% sensitivity, 95% specificity, and 80% power,(±5% precision), including at least 400 HCV RNA-positive and 415 HCV RNA-negative individuals. Sample size calculations were informed by the systematic reviews conducted by Lange *et al.* and Tejada-Strop *et al.* Lange *et al.* reported pooled sensitivity and specificity estimates for HCV RNA testing using DBS of 98% (95% CI: 95–99) and 98% (95% CI: 95–99), respectively [11]. In contrast, Tejada-Strop *et al.* reported sensitivity values as low as 88% [22]. Based on these findings, a conservative assumption of an average sensitivity of 85% and specificity of 95% was applied for the present study. To estimate sensitivity with a 95% confidence interval and a ± 5% margin of error at 80% power, a minimum of 400 participants with detectable HCV RNA was required. For specificity, 415 HCV RNA–negative samples were needed to achieve a 95% confidence interval with a ± 3% margin of error and 80% power.

### Recruitment and consent

Participants were consecutively recruited from clinical databases of "anti-HCV Ab+" group (Cameroon, Georgia and Rwanda) or walk-in clients at study sites (Georgia and Greece). Authorized staff screened anonymized registers to identify eligible individuals. Informed consent was obtained before enrolment, and participants retained the right to withdraw at

any time. In Greece, in addition to inviting PWID visiting the EEEEAIDS to participate, participants were also recruited through a parallel cross-sectional study researching the prevalence of HIV and HCV among PWID in Athens [23].

## Sample collection and processing

Information on consenting participants' demographics, exposure to risk factors, medical history, HIV status, and HBV and HCV serology test results (if known) were documented prior to specimen collection.

At each site, capillary blood obtained by fingerstick was used to prepare six capillary DBS (70µl each) and one capillary PSC (140µl each) per participant. DBS and PSC capillary specimens were collected using graduated EDTA-coated capillary tubes. For DBS, 50–100 µL tubes were used (approximately 70 µL per spot), while 140 µL EDTA capillary tubes were used for PSC collection. The collection procedures were identical for both specimen types. Due to the limited volume of capillary blood that can be taken from one participant within one visit, the sample types evaluated differed by assay platform. Capillary and venous DBS, as well as venous PSC, were assessed only on the cobas 4800, whereas capillary PSC, venous PSC, and venous DBS were evaluated exclusively on the cobas 6800.

All samples were left at room temperature to dry overnight, and then sealed with desiccants, a humidity card and stored at room temperature. Each participant also provided 25 mL of venous blood, which was collected in EDTA (lavender cap) tubes. At each site's laboratory, eleven venous DBS and three venous PSC were prepared from these 25 mL venous blood using automatic pipetting. Within six hours from collection, and following the preparation of the DBS and PSC, venous blood was centrifuged and, subsequently, aliquots of 1 mL of plasma were prepared and stored at −70°C or lower.

## Sample transport

All DBS and PSC were stored at ambient temperature, in order to simulate real-life conditions, and sent to a central laboratory in Australia within 7 days of collection. A temperature monitoring device was included in each shipment, and a humidity card was included in each sealed bag containing DBS and PSC. Plasma aliquots were shipped in dry ice within one month of collection.

Upon arrival at the central laboratory in Australia, DBS and PSC samples were stored at 2–8°C and plasma aliquots were stored at −70°C or lower until testing.

## Sample testing

Within one week of arrival in Australia, capillary and venous DBS and venous PSC were tested for HCV RNA on the cobas 4800 platform at the Royal Hobart Hospital in Hobart (Tasmania), and venous DBS and capillary and venous PSC were tested for HCV RNA on the cobas 6800 platform at the National Reference Serology Laboratory in Melbourne (Victoria).

The tests were performed according to the manufacturers' instructions [24]. At least one negative and one positive manufacturer-supplied control were included in each run of each test.

Samples yielding invalid or discordant results were retested using a second aliquot or, when available, an alternative DBS or PSC specimen from the same participant. Only one repeat test was performed per sample. The reference standard results were available to the performers of the index tests, and vice versa. However, while the index tests were performed within 7 days of arrival in Australia, the reference tests on plasma were performed in batches within more than one month of sample collection.

## Statistical analysis

The primary endpoints of the analysis were sensitivity and specificity, which were calculated by comparing the results of the HCV RNA tests performed on DBS and PSC in both cobas platforms, with the results of HCV RNA tests performed on plasma samples using the same platforms. Sensitivity and specificity were calculated with 95% confidence intervals (95%

CI) using the detection thresholds of the reference assay, with a limit of detection of 15 IU/mL for both cobas 4800 and cobas 6800.

The number of DBS PSC results that were discordant with those of the reference plasma samples, and the results of repeating testing of these samples, were recorded.

To evaluate the quantitative performance of the HCV RNA assays performed on DBS and PSC using the cobas platforms, assay results within the linear range were compared with the results of each assay performed on a paired plasma specimen using $log_{10}$ transformed values. Data were displayed using scatter plots, and analyzed using Deming regression, with the assumption that the measurement errors from both assays were independent and normally distributed. Bland-Altman plots were generated to assess bias and agreement.

Descriptive statistics were used to describe the study population. All analyses were performed using the statistical software R (R Foundation for Statistical Computing, Vienna, Austria, version 3.6). P-values <0.05 were considered statistically significant.

### Management of invalid results

Invalid results were excluded from the sensitivity and specificity analyses. The rate of invalid results was calculated using the indeterminate results in the first testing, by calculating the total numbers of tests without a positive or negative result (result indeterminate) on each cobas platform (S4 Table).

## Results

### Characteristics of the study population

HCV RNA testing was performed on samples provided by 936 consenting participants (Figs 1 and 2). Their median age was 47 years (18–98) and 67% were male (Table 1). Of the total, 84% (n = 720) had an anti-HCV Ab+ profile at the point of enrollment. Almost one in five participants (n = 159, 17%) had received treatment for HCV in the past 12 months. In Georgia and Greece, over 99% of participants (n = 485, 52% of total sample) had a history of non-prescription, injecting drug use. In Cameroon and Rwanda, 0% and 5% (n = 9, 1% of total sample) of participants had a history of drug consumption, respectively.

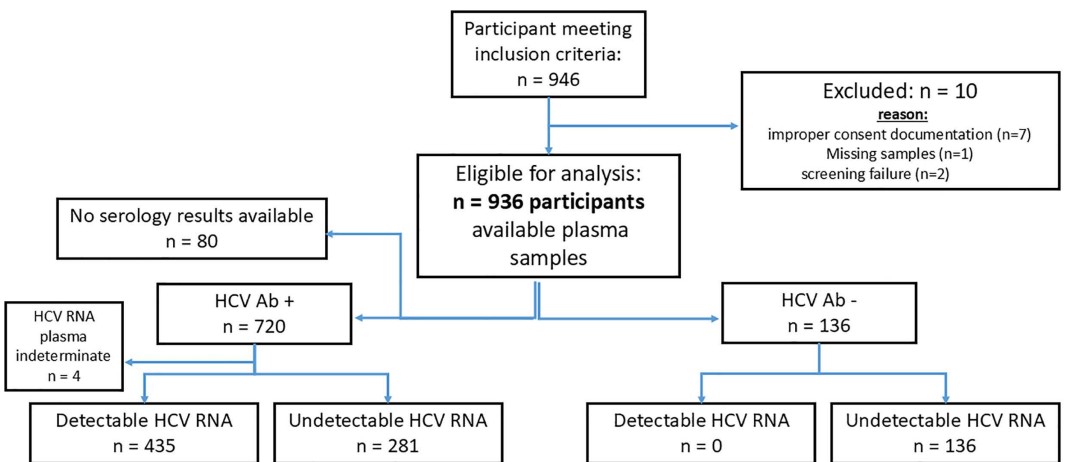

**Fig 1. Study population and study flow diagram.** +, positive; −, negative; Ab, antibody; HCV, hepatitis C virus on the Roche cobas 6800.

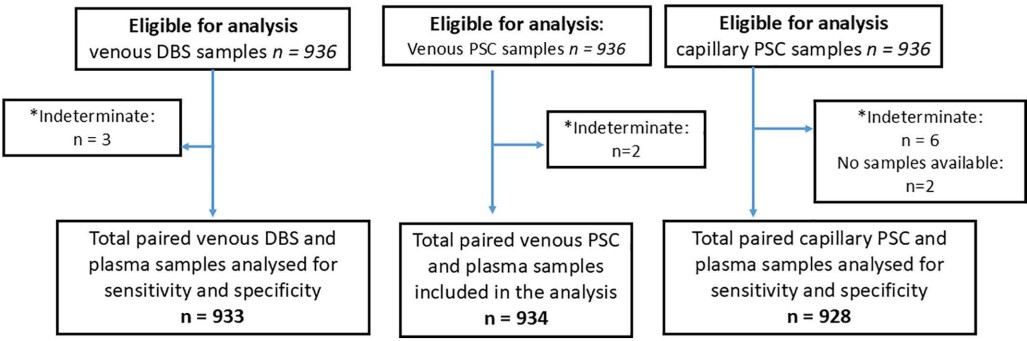

**Fig 2. Eligible capillary, venous plasma separation card and venous dried blood spots, along with paired samples analysed for sensitivity and specificity on the Roche cobas 6800.** The figure displays the number of eligible capillary, venous plasma separation card and venous dried blood spots, as well as the number of paired samples analysed to assess sensitivity and specificity. *Indeterminate on initial testing.

**Table 1. Participants characteristics by site/country.**

|  | Georgia (n=270) | Cameroon (n=249) | Greece (n=218) | Rwanda (n=199) | Total (n=936) |
|---|---|---|---|---|---|
| Sex |  |  |  |  |  |
| Female | 18 (6.7) | 144 (57.8) | 34 (15.4) | 113 (56.8) | 309 (33.0) |
| Male | 252 (93.3) | 105 (42.2) | 184 (84.4) | 86 (43.2) | 627 (67.0) |
| Median age (range), years | 43 (18–69) | 63 (21–98) | 40.5 (19–69) | 55 (22–95) | 47 (18–98) |
| Positive anti-HCV antibody | 181 (67.0) | 249 (100) | 155 (76.4) | 135 (100.0) | 720 (84.0) |
| Persons living with HIV | 3 (1.1) | 2 (0.8) | 17 (7.8) | 15 (7.6) | 37 (4.0) |
| Source of infection |  |  |  |  |  |
| HCV positive mother | 0 (0.0) | 3 (1.2) | 0 (0.0) | 30 (15.1) | 33 (3.5) |
| Injecting non-prescription drug | 269 (99.6) | 0 (0.0) | 216 (99.1) | 9 (4.5) | 494 (52.8) |
| Treated for HCV in past 12 months |  |  |  |  |  |
| No | 270 (100) | 182 (73.1) | 185 (84.9) | 136 (68.7) | 773 (82.7) |
| Yes | 0 (0.0) | 67 (26.9) | 32 (14.7) | 60 (30.3) | 159 (17.0) |
| Detectability of HCV RNA in plasma† |  |  |  |  |  |
| Undetectable | 148 (54.8) | 97 (39.0) | 121 (55.5) | 109 (54.8) | 475 (50.8) |
| Detectable | 122 (45.2) | 152 (61.0) | 97 (44.5) | 89 (44.7) | 460 (49.2) |

Data are number (%) unless otherwise indicated.

† Plasma samples tested using the Roche platforms were used as the clinical reference standard.

HCV, hepatitis C virus; HIV, human immunodeficiency virus.

## Diagnostic accuracy of DBS and PSC samples by platform

The cobas 4800 platform had a sensitivity and specificity of 97% (95% CI: 95 – 98) and 89% (95% CI: 85 – 91) using capillary DBS samples, 96% (95% CI: 94 – 98) and 88% (95% CI: 85 – 90) using venous DBS samples, and 95% (95% CI: 93 – 97) and 100% (95% CI: 99 – 100) using venous PSC samples (**Table 2**). The cobas 6800 platform had a sensitivity and specificity of 97% (95% CI: 95 – 99) and 96% (95% CI: 94 – 97) using venous DBS samples, 97% (95% CI: 95 – 98) and 100% (95% CI: 99 – 100) using venous PSC samples, and 97% (95% CI: 95 – 98)and 100% (95% CI: 99 – 100) using capillary PSC samples (**Table 3**). Site-disaggregated performance of the cobas 4800 and the cobas 6800 is shown in **S2** and **S3 Tables**.

**Table 2. Diagnostic accuracy of capillary DBS, venous DBS and venous PSC for detecting HCV RNA using the Roche HCV nucleic acid assay and Roche cobas 4800 platform.**

| Sample type | Sensitivity (%), (95% CI) | Specificity (%), (95% CI) |
|---|---|---|
| Capillary DBS | 97.2 (95.2 – 98.3) | 88.6 (85.4 – 91.2) |
| Venous DBS | 96.1 (93.9 – 97.5) | 87.8 (84.5 – 90.4) |
| Venous PSC | 95.2 (92.8 – 96.8) | 99.6 (98.5 – 99.9) |

+, positive; −, negative; 95% CI, 95% confidence interval; DBS, dried blood spot; PSC, plasma separation card.

**Table 3. Diagnostic accuracy of venous DBS, capillary PSC and venous PSC for detecting HCV RNA using the Roche HCV nucleic acid assay and the Roche cobas 6800 platform.**

| Sample type | Sensitivity (%) (95% CI) | Specificity (%) (95% CI) |
|---|---|---|
| Venous DBS | 97.3 (95.4 – 98.5) | 95.9 (93.7 – 97.3) |
| Capillary PSC | 96.9 (94.8 – 98.1) | 99.8 (98.8 – 100) |
| Venous PSC | 96.7 (94.6 – 98.0) | 99.8 (98.8 – 100) |

+, positive; −, negative; 95% CI, 95% confidence interval; DBS, dried blood spot; PSC, plasma separation card.

## Repeat testing of discordant results

A total of 154 samples (out of 162 that required re-testing) (**S1 Table**) across four countries underwent repeat testing to investigate discrepancies observed during initial HCV RNA testing. Across all sample types—venous dried blood spots (vDBS), capillary dried blood spots (cDBS), venous plasma separation cards (vPSC), and capillary plasma separation cards (cPSC)—the majority of repeat results were *target not detected* (S1 Table), consistent with prior reports of variability at low viral loads when using alternative specimens such as DBS and PSC.

Among vDBS samples (n = 89) (**S1 Table**), most initial discrepancies involved detected but non-quantifiable results or viral loads below the assay limit of quantification. Upon repeat testing, 93% (82/89) were *target not detected,* with only a small number 2% (2/89) producing a quantifiable viral load. Similarly, cDBS samples (n = 47) showed high concordance on repeat testing: 85% (40/47) were t*arget not detected*, confirming that many initial low-positive results likely reflected borderline viral levels near the assay's detection threshold.

PSC samples showed similar trends but with greater variability. Repeat testing of venous PSC (vPSC) samples (n = 18) also reflected low reproducibility of initial borderline results. Although a small number of prior results were quantifiable (22–16,500 IU/mL), more than half (56%) were *Target Not Detected* on repeat, while 22% remained *<titer* and 22% produced quantifiable viral loads. For cPSC (n = 8), no repeat samples were available, limiting interpretation.

Overall, these results reinforce that discrepant low-positive results from DBS and PSC specimens are frequently not reproducible upon repeat testing. This pattern is consistent with known assay performance characteristics at low viral

loads and aligns with prior literature demonstrating reduced reproducibility around the limit of quantification. These findings support cautious interpretation of isolated low-positive results when using DBS or PSC in clinical or surveillance settings.

## HCV RNA levels according to sample type on the cobas 4800

On the cobas 4800, median capillary and venous DBS VL (approximately 4.1 log IU/mL; **S1B and S1C Fig**) and median venous PSC VL (approximately 4.3 log IU/mL; **S2A Fig**) were lower than the median plasma VL (5.9 log IU/mL for all sites; **S1A Fig**).

The mean difference between VL measured on venous DBS samples and plasma samples using the cobas 4800 platform was > 0.5 log; however, 95.1% of capillary DBS samples had a VL within 1.96 standard deviations (SD) of the plasma VL (**Fig 3C**). There was a strong positive linear correlation between capillary DBS VL and plasma VL (r2: 0.95, p < 0.001; **Fig 3D**). Similarly, there was a strong linear correlation between the VL measured on venous PSC and VL measured on plasma samples (r2: 0.93, p < 0.001, **Fig 4B**).

## HCV RNA levels according to sample type on the cobas 6800

On the cobas 6800, the median plasma VL was 6.1 log IU/mL for all sites but Greece, where the median plasma VL was 5.9 log IU/mL (**S3A Fig**). The median venous DBS VL was 4.3 log IU/mL, also for all sites but Greece, where a slightly

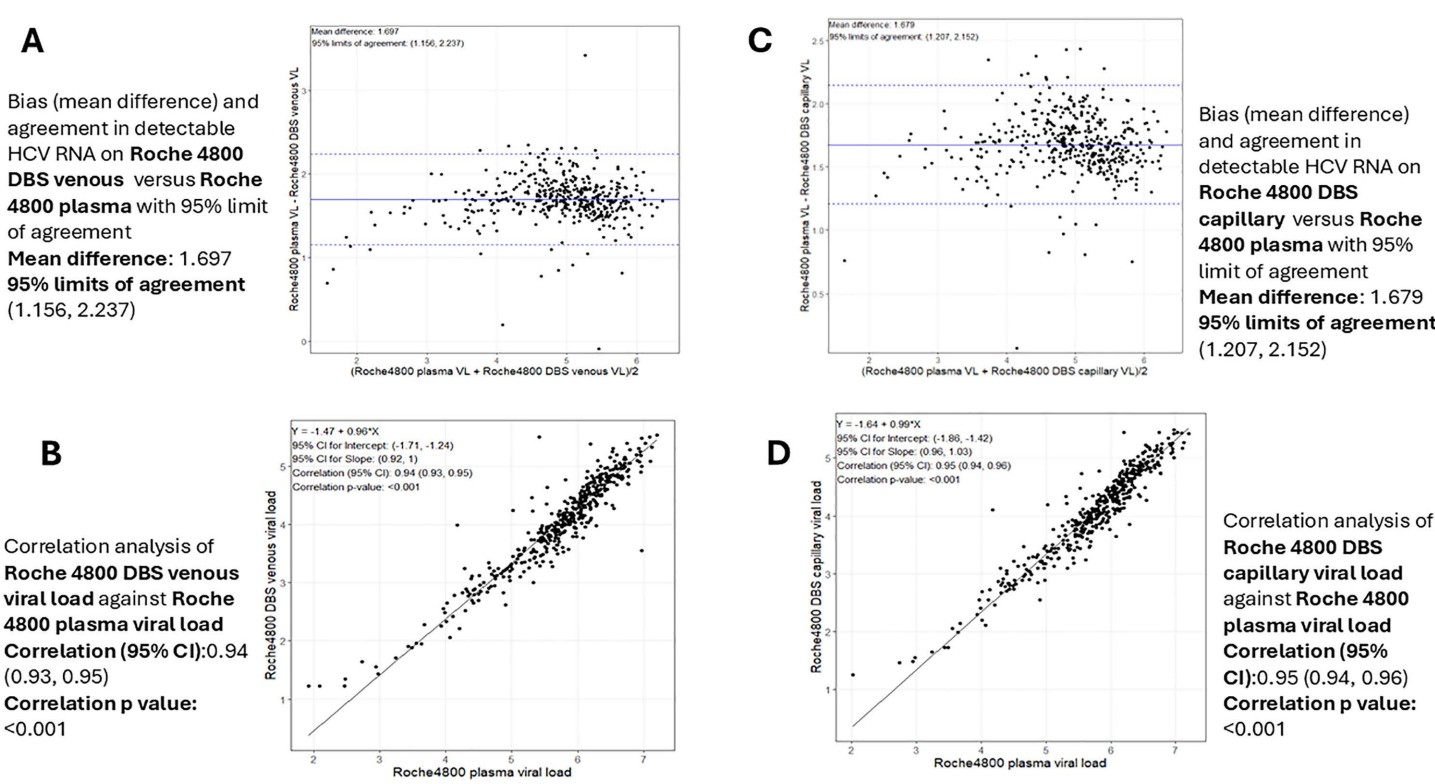

**Fig 3. Bland-Altman analysis and correlation between HCV RNA levels in capillary DBS and venous DBS and plasma samples, measured using the cobas 4800 platform. (A)** Bland-Altman analysis on venous DBS; **(B)** Linear correlation on venous DBS; **(C)** Bland-Altman analysis on capillary DBS; **(D)** Linear correlation on capillary DBS. DBS, dried blood spot; HCV, hepatitis C virus; VL, viral load.

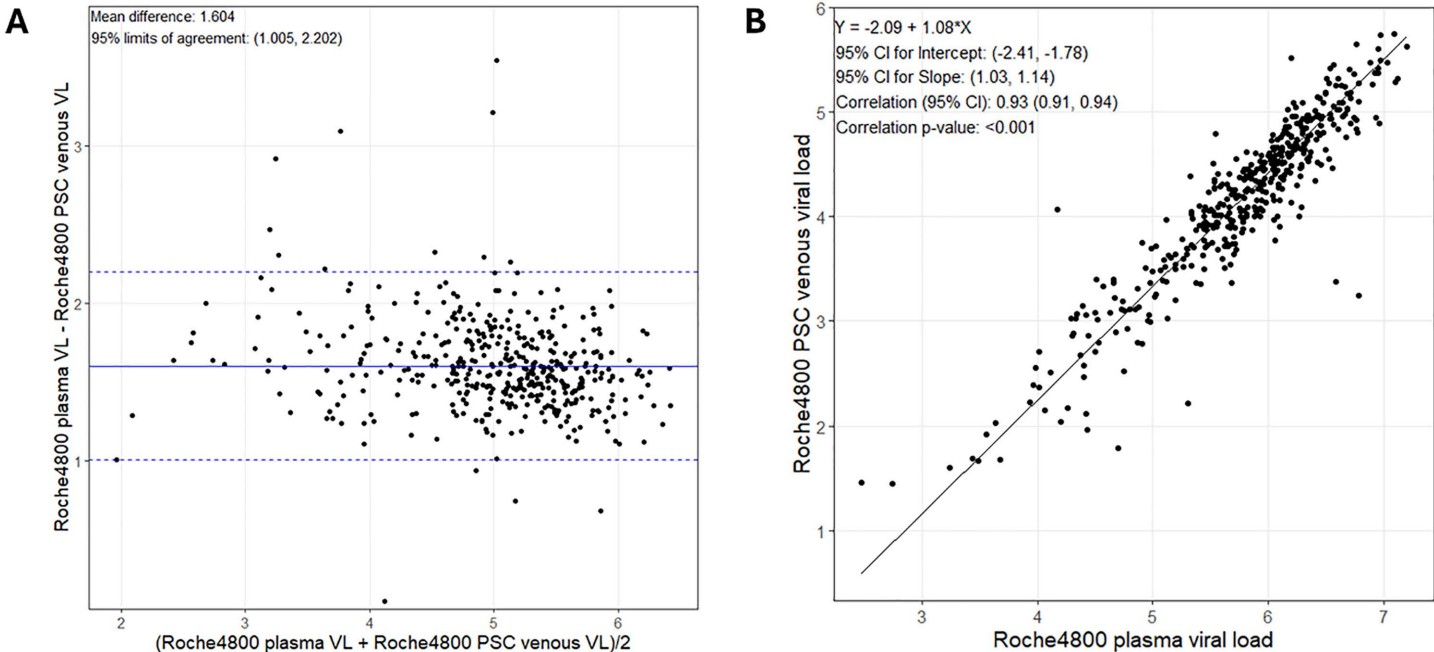

**Fig 4. Bland-Altman analysis and correlation between HCV RNA levels in venous PSC and plasma samples, measured using the Roche cobas 4800 platform. (A)** Bland-Altman analysis; **(B)** Linear correlation. PSC, plasma separation cards; HCV, hepatitis C virus; VL, viral load.

lower median VL of 4.2 log IU/mL was detected (**S3C Fig**). The median capillary and venous PSC VL were also lower than the median plasma VL, both at approximately 4.5 IU/mL log (**S4A and S4B Fig**).

The mean difference between VL measured on venous DBS and plasma samples using the cobas 6800 platform was > 0.5 log; however, 96.5% of venous DBS had a VL within 1.96 SD of the plasma VL (**Fig 5A**). There was a strong positive linear correlation between the VL measured on venous DBS and VL measured on plasma samples (r2: 0.94, p < 0.001, **Fig 5B**).

### Invalid rates

Invalid rates were low (<2.0%) with both DBS and PSC and on both cobas platforms (**S4 Table**).

### Discussion

This study represents, to the best of our knowledge, the first large multicentre evaluation of the clinical diagnostic accuracy of DBS and PSC for HCV RNA detection in real-world settings. It is also the first multicentre validation of DBS and PSC-based HCV RNA testing on the Roche cobas 4800 and 6800 platforms conducted in routine clinical environments in LMICs, capturing real-world operational challenges such as temperature fluctuations, prolonged transport times, and manual punching procedures. The study demonstrated diagnostic accuracy of the Roche cobas 4800 platform (i.e., sensitivity and specificity of 97% (95% CI: 95 – 98) and 89% (95% CI: 85 – 91) using capillary DBS, 96% (95% CI: 94 – 98) and 88% (95% CI: 85 – 90) using venous DBS, and 95% (95% CI: 93 – 97) and 100% (95% CI: 99 – 100), using venous PSC) and of the Roche cobas 6800 platform (sensitivity and specificity of 97% (95% CI: 95 – 98) and 96% (95% CI: 94 – 97) using venous DBS, 97% (95% CI: 95 – 98) and 100% (95% CI: 99 – 100) using venous PSC, and 97% (95% CI: 95 – 98) and 100% (95% CI: 99 – 100) using capillary PSC). These findings suggest that using DBS and the Roche cobas PSC as sampling approaches holds promise as a substitute for plasma sampling in laboratory and clinical resource-constrained

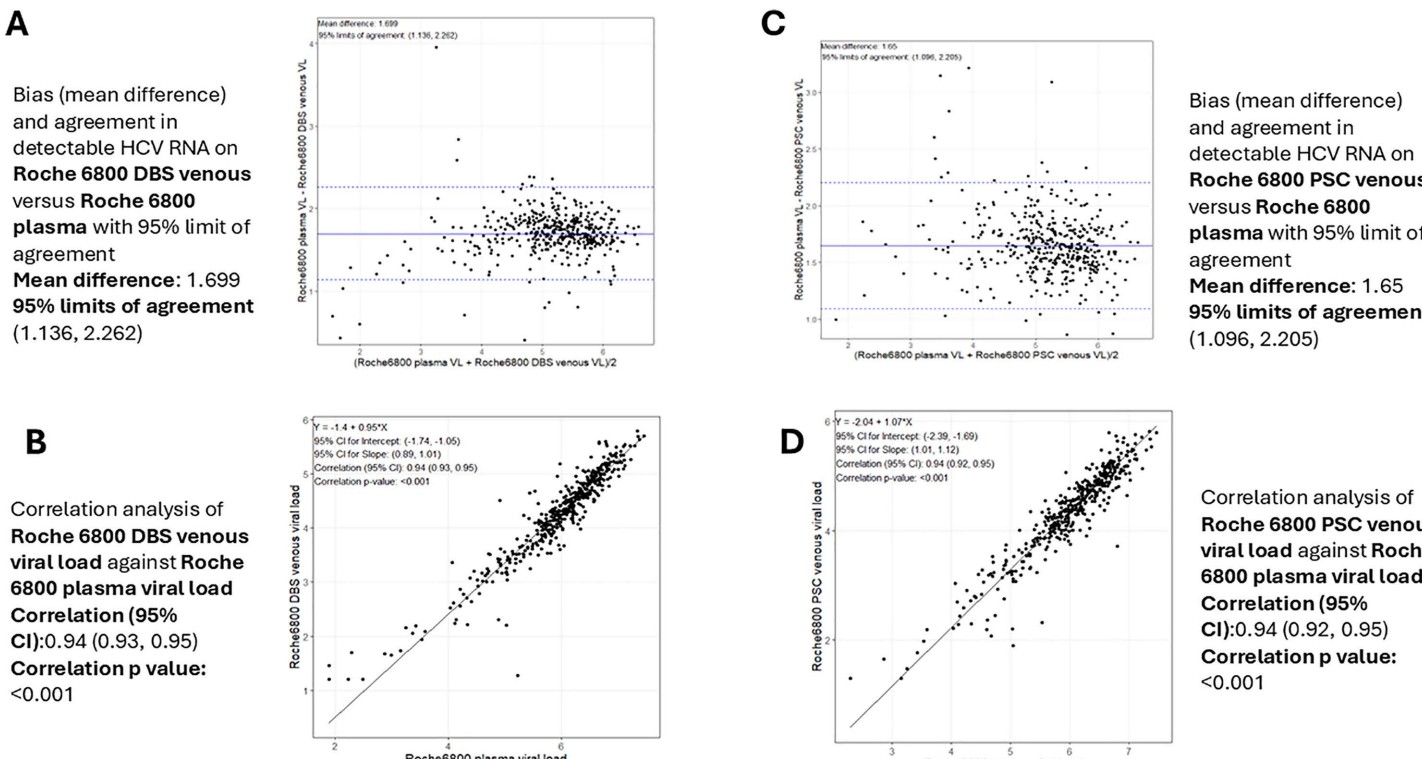

**Fig 5. Bland-Altman analysis and correlation between HCV RNA levels in venous DBS and venous PSC and in plasma samples measured using the cobas 6800 platform. (A)** Bland-Altman analysis on venous DBS; **(B)** Linear correlation on venous DBS; **(C)** Bland-Altman analysis on venous PSC; **(D)** Linear correlation on venous PSC. DBS, dried blood spot; PSC, plasma separation card; HCV, hepatitis C virus; VL, viral load.

settings in LMIC. More often, plasma samples are utilised, which can be challenging to collect, store, transport and handle in low-resource countries. Inclusion of capillary whole blood–based molecular assays aligns with efforts to expand confirmatory testing accessibility in resource-limited or community-based settings. Our findings are similar to those of Vermehren et al. [25], who reported high sensitivity and strong correlation of the cobas 4800 and cobas 6800/8800 tests with other established Roche assays used in clinical practice.

In our study, the cobas 6800 performed relatively well with a high sensitivity and specificity on all sample types tested. However, the cobas 4800 had a lower specificity for capillary DBS (89%; 95% CI: 85 – 91) and venous DBS (88%; 95% CI: 85 – 90) in comparison with PSC and with the cobas 6800. These lower specificities might have been the result of cross-contamination during the preparation of capillary and venous DBS at the sites' laboratories due to the use of perforators (i.e., the DBS were not pre-punched).

Building on Catlett et al. (2022) [14], who reported variable DBS performance at low HCV viral loads, our multicountry, real-world evaluation of the Roche cobas 4800 and 6800 assays confirms non-reproducibility of low-positive DBS results in routine LMIC settings and extends the evidence base through additional validation of plasma separation cards. Across all participating countries, repeat testing of DBS and PSC samples demonstrated a consistent pattern of non-reproducibility among initially low-positive or non-quantifiable results. Collectively, these findings show that most low-level DBS results fail to reproduce on repeat testing, reinforcing known limitations near the limit of detection and underscoring the importance of confirmatory testing when clinically feasible [26–28]. Samples that were initially reported as undetectable but later yielded high quantifiable viral loads (S1 Table) may reflect mislabeling errors that led to discrepancies between plasma

and DBS or PSC results upon repeat testing. Other potential contributing factors for low level of viral load, may include inadequate cleaning of the DBS puncher, which can introduce cross-contamination—though this would not apply to PSC samples, which do not require punching.

Turnaround times between DBS collection and testing (>14 days), combined with exposure to variable temperatures (3.8–55.6°C) during shipment to Australia, may also have negatively affected specificity. Previous studies similarly reported declines in HCV RNA concentrations in DBS stored at ambient temperatures beyond six days [17,18].

On the other hand, using the cobas 4800, venous PSC showed a similar sensitivity (95%; 95% CI: 93 – 97) and a greater specificity (100%; 95% CI: 99 – 100) than capillary DBS (97% (95% CI: 95 – 98) and 89% (95% CI: 85 – 91) and venous DBS (96% (95% CI: 94 – 98) and 88% (95% CI: 85 – 90). This finding is consistent with prior smaller studies suggesting that PSC offers improved sample stability relative to DBS [ 29]. The enhanced stability of PSC likely reflects its design: plasma is separated through a porous membrane and collected on a polyester fleece impregnated with RNA stabilizers, with fewer cellular contaminants than DBS [10,18].

Finally, the strong positive linear correlations of VL measured in DBS and PSC on the cobas 4800 in comparison to plasma samples show that DBS and PSC can be used to quantify HCV VL. However, the somewhat lower VL in DBS and PSC compared with VL in plasma should be considered if DBS or PSC samples are to be used in clinical settings to guide initiation of HCV treatment in HCV-prevalent geographies where scarce health resources for HCV care warrant that clinical decision-making be based on highly-accurate diagnostic approaches. Although DBS is appropriate for genotyping and HCV RNA detection, a systematic review in 2014 already pointed out the need to carry out further investigations to assess the use of DBS for HCV RNA VL measurement [20,30]. Despite our observation with regards to lower VL level in DBS than in PSC, it should be noted that the WHO, in its 2022 updated recommendations on simplified HCV service delivery, indicated that a lower sensitivity level (with a limit of detection of 3000 IU/mL or lower) can be considered to diagnose viremic infection if an assay that can be used at the point of care and is suitable for DBS is able to improve access and/or affordability [1].

DBS holds promise to increase the demand for HCV RNA diagnostics by expanding testing access to rural, remote populations and simplifying sample collection, storage, and transportation [31]. The implementation of DBS is feasible in LMICs due to the relatively simplified sample collection process and minimal training requirements, as has been demonstrated by the use of DBS for early infant HIV diagnosis [32] or for HIV antiretroviral drug resistance [33] various LMICs. DBS can be performed at lower levels of the health system, thereby increasing the potential pool of at-risk persons that can access HCV testing. Nevertheless, as of 2021, the only WHO prequalified platform for dried samples of HCV VL is the Abbott HCV RealTime [31,34,35]. As DBS might play an increasingly large role in HCV RNA testing in LMICs and prequalification is a requirement for some LMICs to access diagnostic equipment, further efforts to build robust evidence on the diagnostic performance of other platforms using DBS are necessary. Additionally, DBS-based HCV antibody testing offers a practical and cost-effective first-line screening strategy, particularly in decentralized or resource-limited settings. Evidence shows high diagnostic performance for anti-HCV detection from DBS, with pooled sensitivity and specificity >95% [11,36]. By performing HCV RNA testing only on DBS samples that are antibody-positive, programmes can substantially reduce molecular testing costs while minimizing patient recall and loss to follow-up [14,37]. This reflex approach streamlines workflows, avoids additional blood draws, and supports scalable HCV elimination efforts across diverse populations.

In addition to using DBS to help increase awareness of HCV status in remote settings, DBS holds potential to increase testing uptake among minoritized, hard-to-reach populations such as PWID and persons who engage in sex work or in same-sex relations, who may see their rights to HCV care jeopardized in countries where they suffer the effects of stigma and of state criminalization. These populations could also see their access to HCV testing improved if the ability to self-sample could be provided. The feasibility of DBS self-sampling in home settings has been demonstrated [38] and it is an approach that – in addition to facilitating sample collection at people's own convenience and in private – could help ensure populations can continue to access HCV testing during future pandemics or health emergency

scenarios when visiting healthcare centers is challenging. DBS self-sampling should be considered as a complementary approach to the delivery of rapid HCV tests for self-testing [39], by organizations providing HCV screening and testing in community-based, mobile units targeting minoritized, hard-to-reach populations [39]. The high diagnostic accuracy of venous PSC tested using the cobas 4800 platform demonstrates the suitability of PSC for HCV RNA testing. However, PSC have higher costs than DBS, and are a commodity licensed by Roche, which will likely determine their utilization in LMICs. Additionally, while the WHO and the European Association for the Study of the Liver (EASL) guidelines support the use of DBS as an alternative to serum or plasma for HCV RNA testing, these organizations have not yet released recommendations for the use of PSC [1,40]. Future socio-economic evaluations and market studies are necessary to advice on PSC production, distribution and utilization methods that could help bring down current costs of PSC. Although the use of dried blood spots (DBS) and plasma separation cards (PSC) remains outside the instructions for use (IFU) of most commercial assays [41–43], the evidence generated through clinical performance evaluations such as this study plays a critical role in bridging the gap between research and routine diagnostic practice. Notably, Roche's PSC has received regulatory approval for use with hepatitis C virus (HCV) RNA and HIV-1 viral load testing, highlighting the potential for broader clinical adoption of these alternative sampling approaches when supported by robust performance data [10,17,18]. Demonstrating robust concordance and acceptable sensitivity and specificity using alternative sample types provides valuable data for manufacturers seeking to extend assay claims and for regulatory authorities considering broader sample-type approval [44,45]. These findings also support advocacy efforts to encourage inclusion of DBS and PSC in kit inserts and regulatory submissions [6,46,47], thereby facilitating their adoption into routine clinical diagnostics. Ultimately, the goal is to integrate these simplified sampling approaches into standard-of-care testing pathways—without relying solely on research frameworks—to enhance diagnostic access and equity in decentralized and resource-limited settings [48].

Despite all the evidence regarding the feasibility and accuracy of DBS for confirmation of chronic HCV infection, their adoption and scale-up by LMICs is slow. The reasons for the dearth of settings with operational experience with DBS is not known. A reason might be that plasma sampling would still be necessary to assess VL levels or to carry out tests to confirm sustained virological response following patients' completion of HCV treatment. Hence, further evaluations are needed to establish whether DBS or PSC samples can be used as an alternative to plasma for VL monitoring to identify treatment curation as well as treatment failure in individuals with low-level viremia.

At the time of writing, and to the best of our knowledge, although these data have been shared with Roche, there is no public indication that they are being considered for regulatory registration purposes. Furthermore, there is currently no indication that the studies will be repeated, despite software updates to the cobas 4800 and 6800 systems, as such efforts would require additional resources. Building on Catlett et al. (2022), who reported variable DBS performance at low HCV viral loads, our multicountry, real-world evaluation of the Roche cobas 4800 and 6800 assays confirms non-reproducibility of low-positive DBS results in routine LMIC settings and extends the evidence base through additional validation of plasma separation cards.

This study has several limitations. First, due to the limited volume of capillary blood that could be collected from a single participant during one visit, the sample types evaluated on each assay differed: capillary DBS were assessed only on the cobas 4800 platform, while capillary PSC were evaluated only on the cobas 6800 platform. Therefore, further studies comparing the performance of both PSC sample types on the cobas 4800 and both DBS types on the cobas 6800 would be valuable. One of the major challenges with DBS testing is occasional discordance at very low viral levels. These assays are highly sensitive and may detect virus that is not clinically relevant. Consequently, some DBS samples that are initially HCV RNA detectable but not quantifiable may test HCV RNA undetectable upon repeat testing. For this reason, additional analyses using the limit of quantification (i.e., HCV RNA detectable and quantifiable) can be informative. Moreover, it is generally recognized that assay performance improves at a threshold of 1,000 IU/mL. Unfortunately, the study was conducted six years ago, and no funds remain to support additional testing or post-hoc analyses. Despite

this limitation, we hope the current findings still provide meaningful evidence to inform the field. Some samples (particularly those from Cameroon) were tested with significant delay due to issues with reagent shipments, which may have influenced quantitative results. The use of a pre-punched filter paper could have reduced the possibility of cross contamination, leading to an underestimation of the rate of false positive results likely to occur in clinical settings. Finally, different types of healthcare workers with varying levels of experience collected the samples at the different sites (e.g., nurses in Georgia and doctors in Greece), which may have affected the quality and amount of capillary blood by fingerstick deposited onto the DBS and PSC spots.

In conclusion, the cobas 4800 and cobas 6800 platforms performed well with DBS and PSC for the detection of HCV RNA and quantification of HCV VL. Specimens collected using DBS and PSC can serve to confirm chronic viremic HCV infection in decentralized, health resource-constrained settings. As DBS and PSC can be prepared following minimal investment in training and equipment, these sampling approaches are a feasible alternative to plasma sampling in LMICs. Utilization of DBS and PSC could impact uptake of HCV screening and accelerate access to direct-acting antivirals for the general population and for minoritized, hard-to-reach populations at increased risk of seeing their rights to HCV screening and care neglected in jurisdictions that criminalize drug consumption, sex work, or same-sex relations.

## Supporting information

**S1 Table. Summary of Repeat Testing Outcomes for Discordant DBS and PSC Samples.** DBS, dried blood spot; PSC, plasma separation card. vDBS, venous dried blood spot; cDBS, capillary dried blood spot; vPSC, venous plasma separation card; cPSC, capillary plasma separation card; Detectable, quantifiable viral load;<titer, detectable, non-quantifiable, i.e., detectable signal below assay limit of quantification; Undetectable, no viral RNA detected. Indeterminate, invalid/error/no result.
(DOCX)

**S2 Table. Diagnostic accuracy of dried blood spot and Plasma Separation Card samples for detecting hepatitis C virus RNA using the Roche cobas 4800 system, according to site.** CI, confidence interval; DBS, dried blood spot; PSC, plasma separation card.
(DOCX)

**S3 Table. Diagnostic accuracy of dried blood spot and Plasma Separation Card samples for detecting hepatitis C virus RNA using the Roche cobas 6800 system, according to site.** CI, confidence interval; DBS, dried blood spot; PSC, plasma separation card.
(DOCX)

**S4 Table. Proportion of invalid results according to assay, sample type, and site.** DBS, dried blood spot; PSC, plasma separation card.
(DOCX)

**S1 Fig. Distribution of hepatitis C virus load tested using the Roche cobas 4800 HCV assay.** (A) Plasma; (B) capillary DBS; (C) venous DBS. The line in the middle of each box represents the median viral load; the top and bottom of each box represent the 75th and 25th centiles, respectively. DBS, dried blood spot.
(TIF)

**S2 Fig. Distribution of hepatitis C virus load tested using the Roche cobas 4800 HCV assay.** (A) venous PSC. The line in the middle of each box represents the median viral load; the top and bottom of each box represent the 75th and 25th centiles, respectively. PSC, plasma separation card; HCV, hepatitis C virus.
(TIF)

**S3 Fig. Distribution of viral load tested using the Roche cobas 6800 HCV assay.** (A) Plasma; (B) capillary PSC; (C) venous DBS. The line in the middle of each box represents the median viral load; the top and bottom of each box represent the 75th and 25th centiles, respectively. DBS, dried blood spot; PSC, plasma separation card; HCV, hepatitis C virus. (TIF)

**S4 Fig. Distribution of hepatitis C virus load tested using the Roche cobas 6800 HCV assay.** (A) capillary PSC; (B) venous PSC. The line in the middle of each box represents the median viral load; the top and bottom of each box represent the 75th and 25th centiles, respectively. Abbreviations: PSC, plasma separation card dried; HCV, hepatitis C virus. (TIF)

## Acknowledgments

We thank the staff in Cameroon, Greece, Georgia, Rwanda and the National Reference Laboratory in Australia who were part of the DBS evaluation study team for conducting recruitment and testing. We also thank Luca Felica, Faustin Gasaza, and Samuel Mukendi from Abbott Molecular Diagnostics for conducting local training in the study sites. We thank Roche Molecular Systems Inc., for donating the test kits and reagents for this study. In addition, we thank Rachel Wright, PhD and Guillermo Z. Martínez-Pérez, PhD for writing assistance, language editing, and proofreading of later drafts.

Dried blood spots (DBS) and plasma separation cards (PSC) offer an alternative to plasma for hepatitis C virus (HCV) RNA testing. This multicenter study confirmed their high diagnostic accuracy on Roche cobas platforms, supporting their use in expanding HCV testing access.

## Author contributions

**Conceptualization:** Elena Ivanova Reipold.

**Data curation:** Aurélien Macé.

**Formal analysis:** Maxwell Chirehwa.

**Investigation:** Marie Amougou-Atsama, Panagiotis Iliopoulos, Jean-Claude Mugisha, Nino Berishvili, Manana Sologashvili, Emmanuel Fajardo, Richard Njuoum, Angelos Hatzakis, Jules Kabahizi, Claude Mambo Muvunyi, Penny Buxton, Sadaf Mohiuddin, Maia Alkhazashvili.

**Methodology:** Francois Lamoury.

**Writing – original draft:** Agnes Malobela.

**Writing – review & editing:** Agnes Malobela, Elena Ivanova Reipold.

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
