## [Decision Letter · Decision Letter 0]

29 Oct 2025

PGPH-D-25-01945

Sensitivity and Specificity of Dried Blood Spot and Plasma Separation Card Samples for Hepatitis C Virus RNA Testing

Dear Dr. Malobela,

Thank you for submitting your manuscript to PLOS Global Public Health. After careful consideration, we feel that it has merit but does not fully meet PLOS Global Public Health’s publication criteria as it currently stands. Therefore, we invite you to submit a revised version of the manuscript that addresses the points raised during the review process.

We look forward to receiving your revised manuscript.

Kind regards,

Guillaume Fontaine, PhD, RN

Academic Editor

Journal Requirements:

1. Please send a completed 'Competing Interests' statement, including any COIs declared by your co-authors. If you have no competing interests to declare, please state "The authors have declared that no competing interests exist". Otherwise please declare all competing interests beginning with the statement "I have read the journal's policy and the authors of this manuscript have the following competing interests:"

1. Please clarify all sources of funding (financial or material support) for your study. List the grants (with grant number) or organizations (with url) that supported your study, including funding received from your institution.

2. State the initials, alongside each funding source, of each author to receive each grant.

3. State what role the funders took in the study. If the funders had no role in your study, please state: “The funders had no role in study design, data collection and analysis, decision to publish, or preparation of the manuscript.”

4. If any authors received a salary from any of your funders, please state which authors and which funders.

3. Please provide separate figure files in .tif or .eps format.

4. We do not publish any copyright or trademark symbols that usually accompany proprietary names, eg (R), (C), or TM  (e.g. next to drug or reagent names). Please remove all instances of trademark/copyright symbols throughout the text, including ® on page 2, 3, 4, 5, 6, 10, 11, 12, 13, 14, 16, 17.

5. We have noticed that you have uploaded Supporting Information files, but you have not included a list of legends. Please add a full list of legends for your Supporting Information files after the references list.

Reviewers' comments:

Reviewer's Responses to Questions

**Comments to the Author**

1. Does this manuscript meet PLOS Global Public Health’s publication criteria?

Reviewer #1: Yes

Reviewer #2: Yes

2. Has the statistical analysis been performed appropriately and rigorously?

Reviewer #1: Yes

Reviewer #2: Yes

3. Have the authors made all data underlying the findings in their manuscript fully available (please refer to the Data Availability Statement at the start of the manuscript PDF file)?

Reviewer #1: Yes

Reviewer #2: Yes

4. Is the manuscript presented in an intelligible fashion and written in standard English?

Reviewer #1: Yes

Reviewer #2: Yes

Reviewer #1: 1. Thank you for the opportunity to review manuscript titled: ‘Sensitivity and specificity of dried blood spot and plasma separated card samples for hepatitis C RNA testing’. The manuscript is well written, provided good background and important data, which will add value to the literature to help highlight the importance of these alternate sample types and the impact these specimens can provide on diagnostics and treatment, particularly in low-to-middle income countries.

2. Reference is made to obtaining an HCV RNA confirmation result from plasma [only] post HCV antibody reactive result. Throughout the manuscript I recommended including the range of HCV RNA methodologies, which also includes capillary whole blood (CEPHEID Genexpert) to inform the reader of the scope of confirmatory options currently available.

3. On line 129, double checking that you wanted to reference ‘HIV’ for high-throughput molecular platforms and not ‘HCV’.

4. From line 131 reference is made to DBS protocols being “off-label” and “regulatory pathways”. Could you include a short summary of how research findings like this helps inform manufacturers of assay performance using sample types not currently approved within their instructions for use (IFU)/kit inserts; and how this could help form an advocacy piece for the manufacture to include DBS and PSC in their product kit inserts for regulatory registration purposes? Make clear the end goal to have these alternate sample types part of routine clinical care without the need for research frameworks to implement.

5. Please provide the reference to the “parallel cross-sectional study researching the prevalence of HIV and HCV among PWID in Athens” on line 163.

6. Was an EDTA transfer pipette used to collect the capillary PSC specimens? If so, please highlight different collection techniques between capillary DBS and PSC for real-world collections.

7. If a result was invalid, did this sample undergo a repeat test if additional sample was available? This testing pathway is unclear to the reader.

Reviewer #2: Overall, this is a very important study and it is great to see these results finally published. There are a number of strengths, including the large sample size, the wide array of different testing options collected from the same participants, and the range of different settings where testing was performed. This work will provide a major contribution to the literature in this area and fills a gap that has not been attended to by the manufacturers.

Major comments

1) Abstract – It is a bit odd that different parameters are presented for the 4800 and 6800 in the abstract. For the 4800, the information on capillary DBS, venous DBS, and venous PSC are presented. However, for the 6800, the estimates for venous DBS, venous PSC, and capillary PSC are presented. It would be useful if the estimates for all four sample types for both assay types were presented in the abstract for more ready comparison.

2) Methods – The section on study population is a bit unclear. The authors provide the definitions for these three groups, but it is not clear as to the exact inclusion criteria for the study. It would be helpful to have a clearer explanation of the inclusion criteria in this section, rather than a description of the different categories. Alternatively, the authors could provide some of the details about sampling and recruitment in the different settings here as this impacts the inclusion of participants. I understand that it is a bit tricky as there is some overlap in the two sections, but I don’t think that the current study population section as written provides a clear description of the inclusion criteria for enrollment.

3) Methods (sample size determination) – It would be helpful to report what the estimated prevalence was to inform these sample size calculations. I imagine that it was not estimated that there was going to be 49% (400 of 815). So, please provide some more information on the assumptions behind the sample size calculations.

4) Methods – Need to mention how many samples required repeat testing in the results.

5) Methods – Apologies if I missed it, but the authors need to explain somewhere why there was information on capillary DBS, venous DBS, and venous PSC for the 4800 and venous DBS, venous PSC, and capillary PSC samples for the 6800. Why was there no capillary PSC for the 4800 and no capillary DBS for the 6800? This needs to be explained.

6) Methods/results – These are incredibly important data. One of the major issues with DBS testing is that sometimes there is discordance at very low levels. These assays are very sensitive (and probably pick-up virus which is not clinically relevant). As such, sometimes there will be DBS samples which are HCV RNA detectable, but not quantifiable that upon repeat testing are HCV RNA undetectable. As such, it has been useful to conduct additional analyses using the limit of quantification (HCV RNA detectable and quantifiable). Further, it is often well-accepted that these assays have better performance at a threshold of 1000 IU/mL. I would like the authors to consider also providing the sensitivity and specificity for these comparisons using a detectable and quantifiable limit and for samples that are > and <1,000 IU/mL. This is important as we know that the sensitivity and specificity for DBS testing is much better when using these thresholds (Catlett, JID 2022). This is such an important study and these data will be helpful for advocating in the use of DBS testing across a range of countries. I would mention in the methods that this was a post-hoc analysis if this analysis was not pre-specified, but it is still incredibly important information. It would be a shame to not have these analyses to inform the field. I think it is unlikely that this type of study will be done again.

7) Results – I think that this section needs to be expanded. I think the results of the initial tests need to be better explained. What proportion were initially HCV detectable and then on re-testing were HCV undetectable (and vice versa)? This is important and partly relates to the previous point about the limit of detection perhaps being too sensitive. It would be helpful to have a table which outlines this information.

8) Results – It would be helpful if the authors provided a table which highlighted the discrepant results. This somewhat relates to the previous comment. It is important to describe what was the cause of the discrepant results. Were these primarily samples that were detectable and not quantifiable and then undetected with laboratory-based testing? It would be very useful to explain the cause of heterogeneity (including with the HCV RNA levels for the discrepant results).

9) Results – It is somewhat hard to read the diagnostic accuracy tables (Tables 2 and 3). It would be much better if these were presented more in line with standard 2 x 2 tables with the corresponding sensitivity and specificity).

10) Discussion – One thing that could be covered in the discussion is the lack of willingness by manufacturers to either use these data or to generate data from their own studies to facilitate the required studies so that DBS can be added as a sample type to their instructions for use. This has been a major barrier to more widespread uptake of DBS testing in many countries.

11) Discussion – One thing that is missing from the discussion is the need to also have assays for HCV antibody testing from DBS. It is a much more cost-effective strategy to first perform HCV antibody testing and then to perform HCV RNA testing on those DBS samples that were HCV antibody positive. It would be good for the authors to discuss this.

12) Discussion – Apologies if I have missed it, but could the authors please provide some thoughts on the next steps forward now that these data are available? Will these data be used by Roche to support listing of DBS as a sample type? I understand that the software for the 4800 and 6800 have been updated, so perhaps these studies would need to be redone in order for them to use the data to update the instructions for use. Also, I think it would be worth commenting on the fact that the PSC is now an approved indication for the Roche platforms, which provides important capacity for this to be used in broader clinical applications globally.

Minor comments

13) Abstract – It would be helpful if the authors included the 95% confidence intervals for the sensitivity and specificity estimates.

14) Introduction – There is a more recent systematic review of dried blood spot testing by Catlett et al (Catlett B, et al JID 2022) that the authors might want to reference for completeness. This review also noted a higher sensitivity and specificity for detection than the study noted in the introduction by Lange et al . I would also note the sensitivity and specificity for greater than the limit of quantification.

15) Introduction (lines 87-88) – I think it might be worth the authors also mentioning that there are very few studies that have included a comparison of DBS and PSC in the same study. This is a major strength of the current study.

16) Results – I would use injecting drug use instead of injectable drug consumption.

17) Results – I would round all the percentages in the tables to the nearest whole number to improve readability.

18) Discussion – I think it would be helpful for the authors to compare their findings to the systematic review by Catlett et al (JID 2022).

**Do you want your identity to be public for this peer review?** For information about this choice, including consent withdrawal, please see our Privacy Policy

Reviewer #1: No

Reviewer #2: **Yes:** Jason Grebely

---

## [Editor Report · Decision Letter 1]

15 Feb 2026

Sensitivity and Specificity of Dried Blood Spot and Plasma Separation Card Samples for Hepatitis C Virus RNA Testing

PGPH-D-25-01945R1

Dear Ms Malobela,

We are pleased to inform you that your manuscript 'Sensitivity and Specificity of Dried Blood Spot and Plasma Separation Card Samples for Hepatitis C Virus RNA Testing' has been provisionally accepted for publication in PLOS Global Public Health.

Best regards,

Guillaume Fontaine, PhD, RN

Academic Editor